# Effect of Thermal and Oxidative Aging on Asphalt Binders Rheology and Chemical Composition

**DOI:** 10.3390/ma13194438

**Published:** 2020-10-06

**Authors:** Ingrid Gabrielle do Nascimento Camargo, Bernhard Hofko, Johannes Mirwald, Hinrich Grothe

**Affiliations:** 1Institute of Transportation, TU Wien, Gusshausstrasse 28/E230-3, 1040 Vienna, Austria; bernhard.hofko@tuwien.ac.at; 2Institute of Materials Chemistry, TU Wien, Getreidemarkt 9/E-165-01-5, 1060 Vienna, Austria; johannes.mirwald@tuwien.ac.at (J.M.); hinrich.grothe@tuwien.ac.at (H.G.)

**Keywords:** thermal aging, oxidative aging, asphalt binders, rheology, chemical composition

## Abstract

Aging of asphalt binders is one of the main causes of its hardening, which negatively affects the cracking and fatigue resistance of asphalt binders. Understanding asphalt aging is crucial to improve the durability of asphalt pavements. In this regard, this study aims at understanding and differentiating the effect of temperature and oxygen uptake on the aging mechanisms of unmodified asphalt binders. For that, four laboratory aging procedures were employed. The two standardized procedures, rolling thin-film oven test (RTFOT) and pressure aging vessel (PAV), were considered to simulate the short-term and long-term aging of the asphalt binders, respectively. In addition, two thin-film aging test procedures, the nitrogen atmosphere oven aging test (NAAT) and ambient atmosphere oven aging test (OAAT) were employed to assess the effect of thermal and oxidative aging on unmodified asphalt binder properties. The NAAT procedure is based on the principle that the inert gas minimizes the oxidative aging. The rheological and chemical characterization showed that the high temperatures considered during the NAAT procedure did not change the properties of the unmodified asphalt binders. Therefore, it can be hypothesized that no significant thermal and oxidative aging was observed during NAAT aging procedure for the considered binders and that oxidative aging is the main cause for the hardening.

## 1. Introduction

Increased awareness of the environmental impacts generated during the service life of road infrastructures supports the incorporation of ecological concepts during planning, design, construction, operation, management, and recycling of pavements. To promote sustainability, road infrastructure should be cost effective, address environmental concerns, and contemplate social concerns without compromising pavement performance [1,2,3]. 

The enhancement of asphalt pavements durability is a promising way to improve its overall sustainability. The increase in the lifespan of pavements may result in the reduction in the number of pavement maintenance intervention, the use of nonrenewable materials, energy consumption, emissions of greenhouse gases, costs, and waste generation [4,5].

Among several factors, such as pavement design, aggregates characteristics, asphalt binder chemical composition, construction practices, traffic and environment conditions, aging is one of the main factors that impacts the durability of asphalt pavements. Since aging leads to an increase in asphalt binder stiffness and brittleness, it contributes to the premature development of fatigue and thermal cracking [6,7]. Therefore, to increase the durability of asphalt pavements, it is necessary to understand the mechanisms and parameters that lead the pavement to deteriorate earlier than designed. Among these mechanisms stands out the hardening of the asphalt binder during its aging since it affects negatively the cracking resistance of asphalt binders.

The hardening undergone by the asphalt binder can be reversible or irreversible. The reversible hardening (physical aging or steric aging) is caused by the reorganization of asphalt binder molecules and/or wax crystallization, which can be reversed by the warming up of the asphalt binder [8,9,10,11]. The irreversible aging (chemical aging) is caused by different processes such as oxidation, polymerization, evaporation of light components, and exudation of oil components from asphalt binder into a mineral aggregate [11,12].

The kinetics of asphalt binder oxidation is an important instrument used to understand the chemical changes during oxidative aging which results in asphalt binder hardening. The study of Petersen and Harnsberger (1998) [13] showed that asphalt oxidation is a dual mechanism, the first period is characterized by a fast reaction period and then followed by the second period related to slower reactions. In the fast reaction (spurt) period, oxygen reacts with perhydroaromatics to form hydroperoxide that can either react with sulfides to form sulfoxides or/and decompose into free radicals and/or undergo aromatization. During the slower reaction period, the reactions occur at a constant rate, resulting mainly in the formation of ketones and sulfoxide functional groups. The ketones formation during the asphalt binder oxidation is correlated to the increase in asphaltenes and its viscosity [13,14]. The concentration of the products formed (mainly ketones and sulfoxides) is dependent on the oxygen concentration, temperature, and sulfur content [15].

The aging of asphalt pavements occurs throughout its service life. During the production of asphalt mixtures, the binder undergoes aging due to the high temperatures utilized in the mixing process. The aging caused by the production of asphalt mixture together with the degradation of asphalt binders during temporary construction, which is related to storage, transport, and laying, is commonly referred to as short-term aging (STA). The main results of STA are the loss of volatiles and fast oxidation of asphalt binders [6,16]. Consequently, long-term aging (LTA) occurs over the road service life due to the interactions of the pavement with reactive oxygen species (ROS) in the atmosphere, moisture, temperature, and UV radiation. The LTA is mainly connected to the incorporation of oxygen, which leads to the formation of several functional groups, increasing the overall polarity of the material [17].

The STA and LTA are simulated by standardized aging procedures commonly referred to as rolling thin-film oven test (RTFOT) and pressure aging vessel (PAV), respectively. These laboratory aging procedures utilize high temperature and/or pressure to significantly increase oxidation rate and hence reduce the time needed to age the asphalt binder to a similar level as in the field [6,16,18]. Moreover, the setup of the conventional aging procedures allows for simulating the thermal-oxidative aging of the asphalt binder during the service life of the pavement.

Given the lack of studies that examine the key impact of the thermal and oxidative aging on the hardening of asphalt binders, this study aims at understanding and distinguishing the effect of thermal and oxidative aging on unmodified asphalt binders using alternative methods of aging. In the context of this paper, thermal aging refers to all mechanisms of aging except for extrinsic oxidation (e.g., volatilization).

The nitrogen atmosphere oven aging test (NAAT) consists of subjecting a thin film of unmodified asphalt binders to heat (163 °C) under an inert gas for 4 h. The inert atmosphere minimized oxidation caused by the presence of oxygen, which allows a more accurate evaluation of the thermal effects on unmodified asphalt binders. Moreover, the rheological and chemical characterization of the asphalt binders at unaged and at different aging conditions indicated that oxidative aging is the main responsible factor of the hardening of the unmodified asphalt binders. 

## 2. Materials and Methods 

### 2.1. Materials

Two unmodified 70/100 penetration graded asphalt binders, which are commonly used in Central and Western Europe for road pavements were used in this study. These asphalt binders were from different suppliers and were named “A” and “B,” respectively. Their physical properties are shown in Table 1:

### 2.2. Aging Procedures

In this study, four distinct laboratory aging procedures were considered to understand the effect of the temperature and oxidation on two different asphalt binders (A and B). The short-term and long-term aging were simulated by standardized procedures, known as rolling thin-film oven test (RTFOT), EN 12607-1 [21] and pressure ageing vessel (PAV), EN 14769 [22], respectively. Additionally, two thin-film aging test procedures were proposed to investigate the effect of thermal and oxidative aging on asphalt binders, in this study. The proposed procedures were called as nitrogen atmosphere oven aging test (NAAT) and ambient atmosphere oven aging test (OAAT). 

To prepare the thin-film asphalt binder samples which were considered in both procedures (NAAT and OAAT), an amount of 28.8 ± 0.5 g of melted unaged asphalt binder was poured into stainless steel pan with a diameter of 140 ± 2 mm. Immediately, after pouring the asphalt binder, the pan was tilted and then placed on a level platform to ensure a uniform film thickness of about 1.90 mm (Figure 1a). 

After the asphalt binder thin-film samples had cooled down to room temperature, they were placed into pan holders (Figure 1b). Two sets of pan holders loaded with five binder samples each were prepared. One set was used on NAAT and the other on OAAT aging procedure.

In the case of NAAT procedure, the pan holder set was placed into a glass vacuum desiccator. Subsequently, the desiccator was closed by sliding motion between the desiccator and its cap. The closure interface between the cap and the desiccator was coated with grease to ensure sufficient sealing. Consecutively, nitrogen was purged through the hole on the top of the cap for 10 min (Figure 1c), and after that, the hole was closed with a glass stopcock to ensure that nitrogen remained in the desiccator (Figure 1d).

The desiccator filled with nitrogen (NAAT) and the other set of pan holders (OAAT) were placed into a universal heating oven (Memmert, natural convection mode) at 163 °C for 4 h. The temperature in the desiccator was monitored in a preliminary test to ensure that both samples experience the same thermal loading. After the NAAT and OAAT aging procedures were finished, the desiccator and the set pan holder sample were removed from the oven. As for the NAAT procedure, it was necessary to open the stopcock to vent the nitrogen from the desiccator before collecting the binder. The asphalt binders obtained in each one of the aging procedures were collected in metallic cans to be used for further rheological and chemical characterization.

It should be highlighted that the NAAT and OAAT aging procedures have been reproduced twice, for each binder (A and B), to assure the reliability of both aging methods. Fourier-transform infrared spectroscopy technique was used to evaluate the repeatability of NAAT and OAAT aging procedures. The comparison of the carbonyl and sulfoxide indices between the results of a given aged binder showed that both proposed aging procedures demonstrated good repeatability.

The suffixes Unaged, NAAT, OAAT, RTFOT, and RTFOT + PAV were added to each binder named “A” or “B” to specify the binder aging state (Table 2).

### 2.3. Rheological Characterization

The rheological characterization of the neat and aged asphalt binders (NAAT, OAAT, RTFOT, and PAV + RTFOT) was performed using a rheometer (Thermo Scientific HAAKE MARS II, Karlsruhe, Germany). The frequency sweep test was conducted from 10 to 82 °C, with an increment of 6 °C in temperature. 

At each tested temperature, the frequency was varied from 0.1 to 10.0 Hz. For temperatures from 10 to 40 °C, the tests were run in parallel plate geometry with a diameter of 8 mm and sample height equal to 2 mm. For higher temperatures (40–82 °C), a parallel plate with 25 mm diameter and sample height of 1 mm was used. The strain amplitude for the tests was limited to 0.01% to guarantee that asphalt binders were tested in the linear viscoelastic (LVE) domain. 

The dynamic shear modulus (|G*|) is equal to the ratio of the stress to the strain amplitude obtained from frequency sweep test results. The higher this parameter is, the higher is the resistance of the material to deformation under load. The phase angle (δ) is the lag between the shear stress and shear strain, and its value can vary from 0° to 90°. The zero value indicates that the analyzed material is completely elastic, and the 90° value is related to its complete viscous behavior [23,24].

The |G*| and δ master curves were constructed based on the time–temperature superposition principle (TTSP) at 25 °C reference temperature. The Williams–Landel–Ferry model [25] was used to obtain the time–temperature shift factors. Finally, the master curves were determined according to the generalized power-law model [26], which was based on the Christensen-Anderson-Marasteanu (CAM) model [27]. 

### 2.4. Fatigue Resistance

The linear amplitude sweep (LAS) test was considered to evaluate the fatigue damage resistance of asphalt binders at different aging levels. The test was carried out according to provisional standard AASHTO TP 101-14 [28]. The test consists of two steps: (i) The frequency sweep test is run to obtain the undamaged material property (α parameter). In this step, a strain amplitude of 0.1% is applied while the frequency varies from 0.2 to 30 Hz. (ii) The amplitude sweep test is carried out on the same sample used in the previous step. The sample is progressively submitted to a linear increase in the strain amplitude from 0% to 30%, while the frequency is kept constant at 10 Hz. 

The analysis of the results was done under the principle of the viscoelastic continuous damage (VECD) theory, which is based on Schapery’s work potential theory [29]. The calculation of the α parameter obtained from the results of the first step is done according to the VECD theory. The B parameter is equal to twice the α value. Based on the α parameter and the results obtained from the second step, it is possible to determine at any time, the damage accumulation due to the degeneration of the material integrity. The damage value in the failure is considered to calculate the A parameter. Based on the determined parameters A and B, the fatigue performance parameter (Nf) is calculated using Equation (1):(1)$$N_{f}=A{\left(\gamma_{\max}\right)}^{-B},$$
where N_f_ is the number of cycles to failure and γ_max_ is the maximum expected binder strain for a given pavement structure. A and B parameters are determined based on VECD. 

In this study, the tests were carried out for unaged and aged binders (NAAT, OAAT, RTFOT, and RTFOT + PAV) at +20 °C. The adopted failure criterion was the maximum value of the shear stress supported by the material during the amplitude sweep test. The fatigue behavior of the asphalt binder was determined using the spreadsheet provided by the Modified Asphalt Research Center (MARC) of the University of Wisconsin–Madison [30].

### 2.5. Fourier-Transform Infrared (FTIR) Spectroscopy

The spectra were recorded using an Attenuated Total Reflectance (ATR) FTIR spectrometer equipped with diamond crystal (Bruker Optics Inc, Model Alpha II, Ettlingen, Germany). Each spectrum was scanned 24 times at a resolution of 4 cm^−1^ and recorded in a range from 4000 to 600 cm^−1^. The test was conducted two times for each binder at all aging states (unaged, NAAT, OAAT, RTFOT and RTFOT + PAV aged).

The ATR-FTIR principle considered to obtain a spectrum is briefly described in the following sentences. When the infrared radiation hits the ATR crystal (material with high refractive index) at an angle of incidence greater than the critical angle, the beam undergoes total internal reflection. In this circumstance, an evanescent wave is created that extends beyond the surface of the crystal and penetrates a few micrometers into the sample adhered to the crystal. Various functional groups within the material can interact with the evanescent wave (change of dipole momentum), resulting in the adsorption of the infrared radiation at a specific wavelength. Finally, the resultant attenuated radiation is directed to the detector and converted via Fourier-transformation into the respective spectrum. The absorption spectrum shows the absorbed intensity of infrared radiation as a function of wavenumbers [31,32,33]. Certain functional groups exhibit characteristic absorptions of infrared radiation [34]. 

The spectrum can be analyzed to identify the structure of chemical compounds presented on the material. The oxidative aging of asphalt binders is usually evaluated based on the regions of absorbance peaks around 1030 and 1700 cm^−1^. Those band regions are related to the sulfoxide and carbonyl functional groups, respectively [16,35,36,37,38,39,40,41,42]. 

The evaluation of the oxygen uptake during the asphalt binder aging was based on the carbonyl and the sulfoxide indices. Both indices were determined according to the integration method considering the normalized spectrum (at band 2919 cm^−1^) and the absolute baseline. Further details of the calculation method used are given in [43]. Carbonyl and sulfoxide indices were calculated according to Equation (2) and Equation (3), respectively, as follows: (2)$$I_{CO}=\frac{\mathrm{Area}\;\mathrm{around}\;1700{\;\mathrm{cm}}^{-1}}{\mathrm{Area}\;\mathrm{around}\;1460{\;\mathrm{cm}}^{-1}}$$
(3)$$I_{SO}=\frac{\mathrm{Area}\;\mathrm{around}\;1030{\;\mathrm{cm}}^{-1}}{\mathrm{Area}\;\mathrm{around}\;1460{\;\mathrm{cm}}^{-1}}.$$

The wavenumbers integration limits considered for the calculation of the carbonyl index and sulfoxide index are given in Table 3:

### 2.6. SARA Fractionation

To investigate their chemical composition, asphalt binders are fractionated considering the principles of solubility and polarity. The separation of the asphalt binder into fractions aims to facilitate the assessment of the chemical changes that the material undergoes due to aging [44]. The acronym SARA (Saturates, Aromatics, Resins, and Asphaltenes) is used to name each fraction of the asphalt binder. The polarity increases from saturates to asphaltenes.

In this study, the asphalt binder fractionation was carried out according to an adapted version of the SARA fractionation methodology developed by Sakib et al. [45], which is mainly divided into two steps. In the first step, the asphaltenes (n-heptane insoluble) are separated from maltenes (n-heptane soluble) by solvent precipitation followed by filtration. The following step consists of fractioning the maltenes into saturates, aromatics, and resins based on elution-adsorption chromatographic principles using solid-phase extraction (SPE). Comparing solid-phase extraction to conventional elution-adsorption chromatography, SPE provides clear cutoff points in the polarity gradient, while chromatography is run in a continuous gradient mode. Using SPE, a clear classification of different polarity groups is achieved, which is advantageous for further analysis. The main difference between the method proposed by Sakib et al. [45] and the method considered in this study is the procedure used to dry the solvent solutions. In the next paragraphs, the methodology adopted in this study is briefly described. Further details can be found at [45,46]. 

For the separation of asphaltenes from maltenes, 400 ± 40 mg of asphalt binder was dissolved in 40 mL of n-heptane (nonpolar solvent), and the solution was stirred for 24 ± 2 h at room temperature (Figure 2a). Then, 10 mL of stirred solution was poured into the syringe (with PTFE syringe filters of 0.2 µm pore size attached at its tips) mounted on ports of a vacuum manifold. The manifold connected to a vacuum source was used to accelerate the filtration process (Figure 2b). Additionally, 5 mL of n-heptane was passed through the same syringe-filter assuring that all the maltene was dissolved. The solution with maltenes was collected in a preweighed glass vial placed in the manifold and positioned just below the port. The precipitated asphaltenes were retained in the filter. The filtering process was repeated three more times, consuming the entire stirred solution.

Two of the four obtained vials were dried until constant weight to determine the mass of maltene in each vial. For that, the vials were placed on a stirring hotplate at 135 °C, and a flow of nitrogen was used to avoid oxidative aging and accelerate solvent evaporation. For safety reasons, the drying process was carried out inside a fume hood. The two vials were weighed, and the maltene content in the vials was measured gravimetrically.

The maltenes percentage in the asphalt binder was obtained based on the mass values of maltene and asphalt binder. The asphaltenes percentage was determined by subtracting the maltene percentage from 100%. The maltene solution in the two remaining vials (15 mL each) was diluted with n-heptane to achieve approximately 3.33 mg/mL maltene/n-heptane (*w*/*v*) ratio solution, which was used further for the maltene fractionation. 

For the maltenes fractionation, SPE cartridge (5 g silica gel with 20 mL capacity and 60 Å pore size) was mounted in one of the ports of the vacuum manifold. Then, 20 mL of n-heptane was poured on the cartridge to prewash it. For the saturates collection, the cartridge was moved to the next port position and 15 mL of the diluted maltene solution was pushed through the cartridge followed by an extra 10 mL of n-heptane. Then, the cartridge was moved again and 25 mL of 80:20 (*v:v*) toluene: n-heptane solution was poured in the cartridge to gather the aromatics. Finally, the cartridge was moved to the last port position, 40 mL of 90:10 (*v:v*) dichloromethane and methanol solution was applied in the cartridge, to obtain the resins fraction (Figure 2c).

The vials with the eluted fractions were dried until constant weight according to the methodology used for drying the maltene solution, previously mentioned. Lastly, the vials were weighed to determine the saturates, aromatics, and resins fractions gravimetrically. In Figure 2d, the SARA fractions obtained after the asphalt binder fractionation are shown.

The n-heptane, toluene, dichloromethane, and methanol solvents used on the asphalt binder fractionation were high-performance liquid chromatography (HPLC) grade.

## 3. Results and Discussion

At least two replicates were run for all rheological (frequency sweep and LAS) and chemical (FTIR-ATR and SARA) characterization tests. In some cases, additional replicates were run to ensure that the coefficient of variation was within 10%. The LAS, FTIR-ATR, and SARA results are given based on the average value. 

### 3.1. Master Curves

Figure 3a–f show the dynamic shear modulus (|G*|) and phase angle (δ) master curves of binders A and B at different aging stages (unaged, NAAT, OAAT, RTFOT, and RTFOT + PAV) and at a reference temperature of 25 °C. Regardless of the asphalt binder aging state, for both binders, the |G*|values increased and the δ values decreased with the increment of reduced frequency.

For binders A and B, it was observed that the |G*| and δ master curves of unaged state almost overlaid the curves of NAAT (Figure 3e), which indicates that the NAAT procedure did not significantly affect the stiffness and viscoelasticity of the asphalt binders. Based on this finding, it can be hypothesized that two events happened during the NAAT aging procedure: (i) absence of oxidation and loss of volatiles and (ii) no relevant thermal aging of the asphalt binders. The nitrogen atmosphere considered during the NAAT procedure may hinder the oxidation of the asphalt binders, even when the high temperature (163 °C) and extended exposure time (4 h) were considered.

The master curves (|G*| and δ) at OAAT and RTFOT aging conditions were comparable (Figure 3f). This was observed for both binders (A and B). The similarity between the OAAT and RTFOT master curves indicated that the aging undergone by each asphalt binder during both aging procedures was very similar. 

According to Table 4, both procedures (RTFOT and OAAT) considered the same temperature during the aging procedure (163 °C), however, each procedure considered specific values of the binder film thickness, procedure duration, and agitation mechanisms. It can be conjectured that the longer test duration considered in OAAT (240 min) when compared to RTFOT (75 min) was enough to compensate for the effect of higher thickness of the asphalt binder film and the absence of continuous movement during the OAAT aging procedure. 

Moreover, the similarity between the OAAT and RTFOT master curves reinforces the hypothesis that the nitrogen considered in the NAAT aging procedure has inhibited the reaction of asphalt binders with oxygen, given that NAAT and OAAT considered similar aging procedures methodology (the only difference is that NAAT considers an inert atmosphere while OAAT considers an oxidative environment). Consequently, the inert atmosphere of nitrogen has prevented the binders from incorporating oxygen and subsequent physical hardening. In other words, the reaction of the asphalt binders with oxygen may be one of the main reasons for the stiffening of the material during OAAT and RTFOT aging procedures. 

The RTFOT + PAV aging procedure led to an upward shift in |G*| master curves and downward shift in δ curves, in the overall frequency range (Figure 3f)). This is expected because LTA results in the increment of the stiffness and elastic performance of asphalt binders. Comparing the RTFOT + PAV phase angle master curve of both binders (Figure 3f), a different aging pattern is observed for each binder, indicating that LTA aging affected the rheology of the binder A and B in different ways. The increase in elasticity (decreasing in δ values) due to the LTA aging was more evidenced at low and intermediate frequency ranges showing that binder A may present higher resistance to fatigue and rutting when compared to binder B.

To assess the mass change during NAAT and OAAT aging procedure, the mass change rate (MCR) was calculated in accordance with Equation (4). The MCR values regarding asphalt binders A and B are presented in Figure 4a,b, respectively.
(4)$$MCR=\left(\frac{\left(M_{1}-M_{0}\right)}{M_{0}}\right)\times 100,$$
where M_1_ and M_0_ are the mass of the binders after and before aging, respectively.

For both asphalt binders (A and B), it was observed a decrease in the mass of the asphalt binder after the NAAT. This finding indicates that volatile light components that are present in the fresh asphalt binders are evaporating solely due to the high temperatures (163 °C). On the other hand, the hypothesis that no relevant thermal aging of the asphalt binders seems to be reasonable since the rheological results showed that the volatilization of lighter compounds during NAAT did not affect the stiffness of the material. 

In addition, the higher decrease in the mass of the asphalt binders after the NAAT when compared to the decrease after OAAT aging procedure indicates that the loss of mass due to the volatilization of lighter compounds was offset by the oxidation of the material.

In Figure 5, the inverse of the log crossover modulus (1/log|Gc*|) for binder A and binder B is shown. The cross modulus |Gc*|is used to quantify the aging susceptibility of asphalt binders. It corresponds to the value of the modulus when the storage master curve and the loss modulus master curves cross each other. In other words, it is the value of modulus when the phase angle is equal to 45° [47,48]. In some studies, it was found an asphalt dependent correlation between 1/log|Gc*| and carbonyl content. It is expected that the higher this parameter, the greater the oxidation undergone by the material [47,49].

According to Figure 5, the binders in the unaged, RTFOT, and OAAT conditions presented similar values of 1/log|Gc*|, whereas the binders in the conditions NAAT and RTFOT + PAV presented the lowest and highest values of 1/log|Gc*|, respectively. The results obtained suggest that 1/log|Gc*| may not be a suitable parameter to differentiate unaged from RTFOT aged samples. On the other hand, this parameter was effective in differentiating RTFOT + PAV from other binders. In addition, the lower values observed for NAAT binders when compared to unaged binders reinforce the hypothesis of the absence of oxidation and thermal aging during NAAT aging procedure. 

### 3.2. LAS Test 

Table 5 presents the average values of LAS parameters (A and B) obtained from the VECD analysis. The A parameter is correlated with the fatigue life of the material while the parameter B indicates the material sensitivity to an applied shear strain [50,51,52,53,54,55]. When the values of LAS parameters of binders A and B are compared for the same aging state, it was observed that binder A presented a higher absolute value of the parameters, which indicates its superior fatigue performance.

Figure 6 shows the variation of the shear stress due to the progressive linear increase in shear strain during the amplitude sweep test. For the same aging state, binder B presented higher values of peak shear stress and lower values of shear strain at peak compared with binder A. Therefore, based on the obtained results, it can be hypothesized that binder B is more brittle and more prone to fatigue than binder A [44,56].

The fatigue curves of binder A and B are presented in Figure 7. For the same aging state, binder A presented higher fatigue life than binder B. This was expected based on the higher elasticity (Figure 3e,f) and the higher ductility (Figure 6) observed in binder A. 

For both asphalt binders, it was observed that the fatigue curve of the unaged binder overlaps with the NAAT binder curve. The result was in accordance with the rheological analysis, which indicated that the inert atmosphere did not impact significantly the properties of the binders. Thus, no negative thermal aging effect can be observed on NAAT binders. In addition, the hypothesis that OAAT and RTFOT aging procedure age the asphalt binders in a similar way was also checked during the LAS test since similarity was observed between the fatigue behavior of the binders aged by OAAT and RTFOT procedure.

When high deformations were applied, it was observed that aging negatively affected the fatigue life of the material. However, at low levels of deformation (up to 10%), it was found in this study as well as in other studies [57,58,59] that aging can be beneficial for the fatigue life of asphalt binders. Therefore, the prediction of pavement fatigue performance based on LAS results should be done cautiously, given that the study of Bennert and Maher [60] observed the fatigue behavior obtained through the LAS test was not in accordance with the fatigue cracking observed in the field.

### 3.3. FTIR

Figure 8a–c displays the FTIR absorbance spectra for the binders considered in this study.

Usually, the effect of the aging is observed on band areas around the peaks at 1700, 1600, and 1030 cm^−1^, respectively, related to the carbonyl, aromatic, and sulfoxide chemical compounds. The increment of carbonyl and sulfoxide content is due to oxygen incorporation in the material [61,62,63,64]. The aromatization is correlated to the dehydrogenation of naphthenic rings, dimerization process, cyclization of alkyl chain, and aromatization of the perhydroaromatic ring [17,65,66].

For both asphalt binders (A and B), the detailed visual analysis indicated that exposure to nitrogen during NAAT did not result in the formation of nitrogen-containing compounds. However, future chemical analyzes must be carried out to ensure that nitrogen did not react with asphalt binders during NAAT aging procedure.

The summary of the main functional groups found in both binders is listed in Table 6. Both unaged asphalt binders (A and B) show bands assigned to aliphatic (2919, 2850, 1456, 1375, and 721 cm^−1^), aromatic (1600, 865, 810, and 745 cm^−1^), and sulfoxide (1030 cm^−1^) functional groups. The carbonyl functional (1700 cm^−1^) was mainly formed after aging (OAAT, RTFOT, and RTFOT + PAV). 

In Figure 9, the values of carbonyl and sulfoxide indices are presented. The quantitative analysis showed that NAAT and unaged binders presented similar values of carbonyl and sulfoxide indices. The similarity between NAAT and unaged binders indices emphasizes the hypothesis that the inert atmosphere considered in the NAAT procedure was effective to diminish the oxidation of the asphalt binders (A and B) significantly and that no relevant thermal aging could be observed for these binder samples. 

The carbonyl and sulfoxide indices of the OAAT and RTFOT binders were similar to one another and lower than the values obtained for RFTFOT + PAV binders. The possible explanations for the high values of carbonyl and sulfoxide indices observed for RTFOT + PAV binders are the use of a preaged RTFOT binder as a starting point and the application of high-pressure oxidation. The increase in pressure results in greater oxidation of the binder when the same temperature is considered [67]. 

In short, formation of both carbonyl and sulfoxide compounds was related to the rise of the oxidation severity brought by a specific aging procedure. Another important point to be highlighted is that binder A showed higher values of carbonyl indices than binder B, for the same aging state; this finding was in accordance with the drawn prediction based on 1/log|Gc*| parameter. 

### 3.4. SARA

Asphalt binder aging alters their composition and chemical structure [68]. The change in the asphalt composition may result in a decrease in the amount of less polar compounds and increase in the amount of polar compounds. In Figure 10, the change in chemical composition caused by a specific type of aging procedure is shown. As expected, for both binders, the oxidative aging procedures (OAAT; RTFOT, and RTFOT + PAV) led to the decrease in aromatics and the increase in asphaltenes and resins; both components present high polarity. The change of saturates content was almost negligible. These findings are consistent with the results mentioned in other studies [69,70,71]. 

The higher stiffness of binder A when compared to binder B, especially after PAV aging procedure, should not be solely correlated to their asphaltene content [58,72]. Instead, the asphalt binder stiffness must be assessed from the total sum of the resins and asphaltenes fractions. The high polarity of asphaltenes and resins results in strong intermolecular attraction force between polar molecules. The higher the attraction between the molecules, higher is the stiffness of the material. Similarly, the increase in asphaltenes and resins during the aging of the asphalt binder favors the agglomeration of the molecules, which can cause hardening of the asphalt binders [42,65,68,72,73,74,75,76].

Additionally, the comparison between the fractions of the unaged binder to the RTFOT + PAV binder showed that both binders (A and B) presented a similar percentage variation for the aromatic and asphaltenes fractions due to the LTA. For both binders, the percentage of reduction in aromatics was close to 23%, while the percentage of increase in asphaltenes was approximately 38%. The same was not observed for resins fractions, indicating that aromatics may be the main asphaltenes-forming agents during the oxidation of asphalt binders, which has also been mentioned by Qin et al. [77].

For asphalt binders A and B, the values of each of the SARA fractions observed for the binders at unaged condition were very similar to those observed for binders at NAAT aged condition. More precisely, the percentage difference between unaged and NAAT binder was less than 6%, for a given SARA fraction. The results are in accordance with the hypotheses made on the rheology and FTIR spectroscopy analysis regarding the effectiveness of the inert atmosphere (considered on NAAT aging procedure) on preventing changes in the material properties of both binders (A and B).

The micelle theory is usually considered to illustrate the colloidal molecular structure of asphalt binders. According to this theory, the micelles formed by the interaction of highly polar fractions (asphaltene and resins) are dispersed within a low polar maltene phase. The compatibility between the maltene and micelle is possible because the micelles are surrounded by a shell with a continuous polarity gradient. With oxidative aging, the chemical compositional of the shell is altered, which affects the stability of the material [2].

The Gaestel Index (Ic) is usually used to correlate the oxidative aging and the colloidal stability of the asphalt [78]. The Ic is defined as the ratio of the sum of percentage values of saturates and asphaltenes to the sum of percentage values of resins and aromatics [79]. The increase in the value of this index is usually correlated to the decrease in stability of the colloidal system [80,81].

In Figure 11, the values of Ic different aging conditions are presented. For both asphalt binders (A and B), the values of Ic were close to 0.70, which indicated that both asphalt binders have a gel–sol transition behavior [82]. However, it should be noted that in the present study, the SARA fractionation was based on the methodology developed by Sakib et al. (2019) using solid-phase extraction (discontinuous solvent gradient) instead of column chromatography (continuous solvent gradient) [45]. The pore size and the material of the filter used in this methodology lead to a higher yield of asphaltenes when compared to the values obtained by the ASTM D4124 standard. Therefore, it is expected that the Ic values calculated from the method proposed by Sakib et al. (2019) will be higher than those mentioned in other studies [45].

In addition, it is noticed that binder B is more stable in terms of colloidal stability than binder A. This can be explained by its lower asphaltene content. Besides, the higher value of Ic for PAV asphalt binders indicates that the increase in asphaltene content negatively affects the stability of binder A and B.

## 4. Conclusions

In this study, the applicability of alternative laboratory aging methods was evaluated. The proposed methods aim at improving the assessment of the impacts of thermal and oxidative aging on the asphalt binder. The asphalt binders aged by the proposed methods and by RTFOT and PAV aging procedures were evaluated considering rheological and chemical characterization techniques. Based on the obtained test results, conclusions can be drawn as follows:
The unaged binder and the binder aged by NAAT showed no significant difference in terms of the values of dynamic shear modulus, phase angle, fatigue life, carbonyl index, sulfoxide index, and SARA fractions. These observations were verified for both asphalt binders (A and B). The findings indicate that the proposed NAAT procedure was effective in minimizing the impact of oxidative aging (even when they were subjected to high temperatures and for a prolonged period) and consequently prevented the binders from stiffening. In other words, no relevant thermal aging could be observed for either of the binders after NAAT procedure.The asphalt binders aged by RTFOT and OAAT aging procedures showed similar rheological and chemical results.In general, RTFOT + PAV aged binder present the highest values of stiffness and elasticity. However, the increment in elasticity due to the LTA was more evident for binder A than binder B, indicating its better fatigue and rutting performance when compared to binder B.Binder A presented a greater content of polar fractions (resins and asphaltenes) compared to binder B. The greater polarity results in a better association between polar molecules, thus higher stiffness when compared to binder B can be explained.The results of the rheological and chemical characterization showed that the NAAT aging procedure proposed in this study is effective to hinder the oxidation of the asphalt binders. This indicates that the NAAT method is a valuable tool to assess the individual effect that temperature and oxidation have on the aging of asphalt binders.

In summary, the results of the rheological and chemical characterization from samples aged by NAAT procedure showed that the impact of thermal aging (carried out in the absence of oxidation), i.e., mostly driven by volatilization, was negligible. Moreover, the results obtained for both asphalt binders (A and B) indicate that oxidation is the main factor for the hardening of the asphalt binders. It is expected that the conclusions of this research will improve the understanding of the mechanisms that contribute to the aging of asphalt binders. 

## Figures and Tables

**Figure 1 materials-13-04438-f001:**
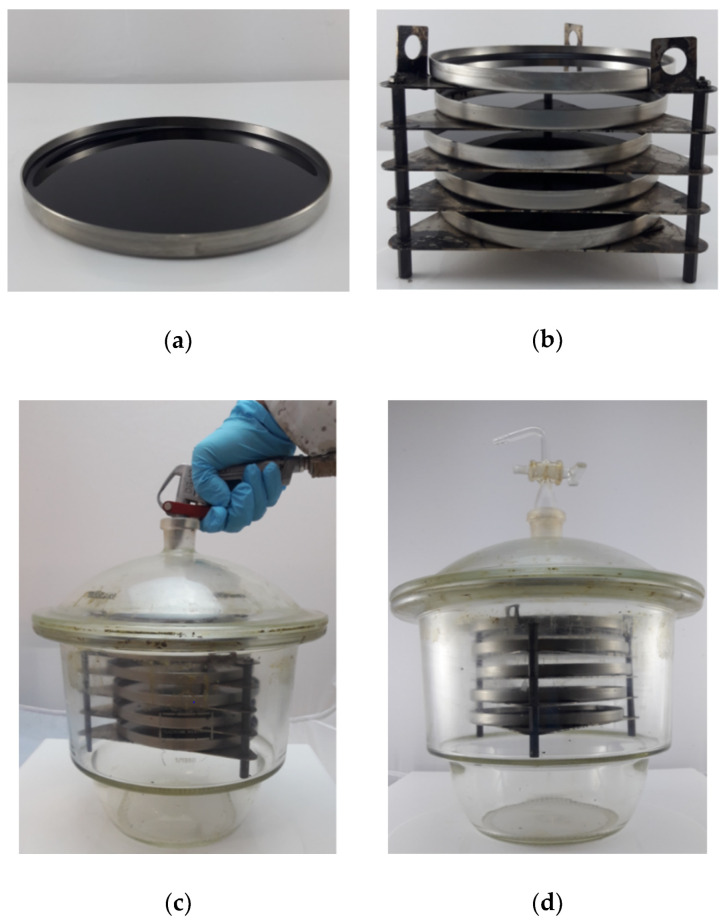
Sample preparation and nitrogen atmosphere oven aging test (NAAT) procedure: (**a**) example of thin-film sample considered in NAAT and ambient atmosphere oven aging test (OAAT) aging procedures, (**b**) set of holder pan filled with thin-film asphalt binder samples, (**c**) the desiccator is filled with nitrogen gas, and (**d**) desiccator filled with nitrogen.

**Figure 2 materials-13-04438-f002:**
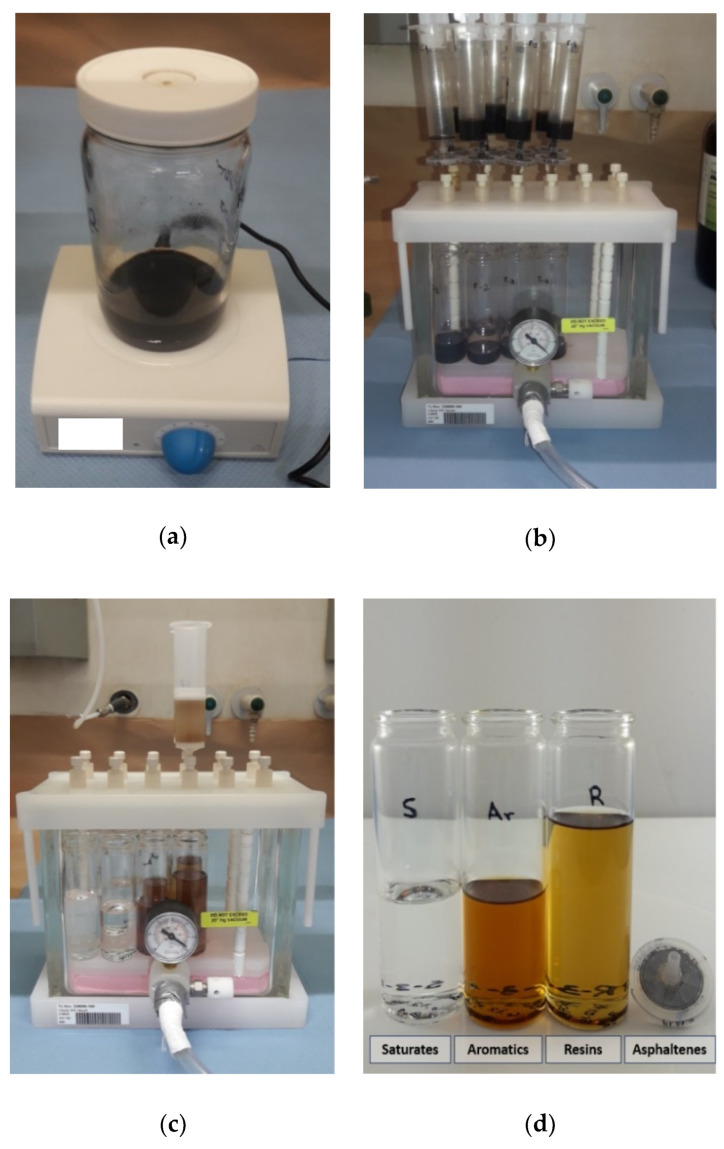
Saturates, Aromatics, Resins, and Asphaltenes (SARA) fractionation: (**a**) asphalt binder dissolution, (**b**) filtration of the solution, (**c**) maltene fractionation, and (**d**) SARA fractions.

**Figure 3 materials-13-04438-f003:**
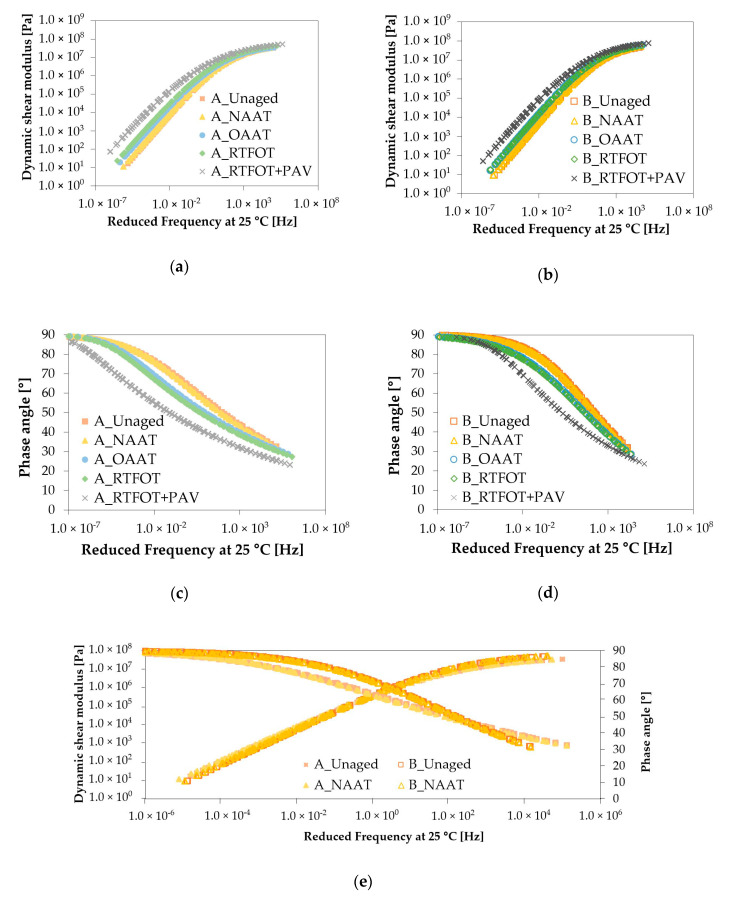
Master curves: (**a**) |G*| master curves of binder A, (**b**) |G*| master curves of binder B, (**c**) δ master curves of binder A, (**d**) δ master curves of binder B, (**e**) |G*|and δ master curves of binder A and B (at NAAT and Unaged conditions), and (**f**) |G*|and δ master curves of binder A and B (at OAAT, rolling thin-film oven test (RTFOT), and RTFOT + pressure aging vessel (PAV) conditions).

**Figure 4 materials-13-04438-f004:**
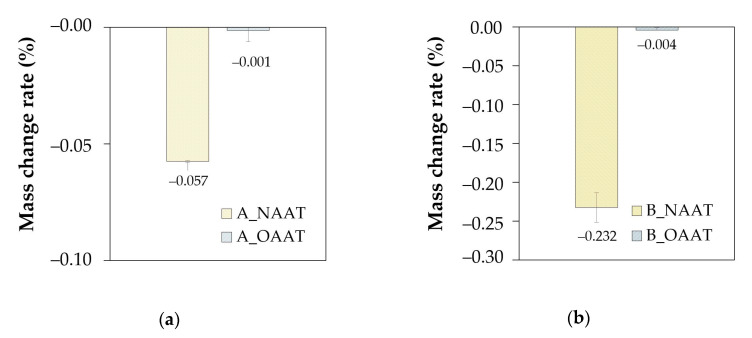
Mass change rate (after NAAT and OAAT aging procedure): (**a**) binder A; (**b**) binder B.

**Figure 5 materials-13-04438-f005:**
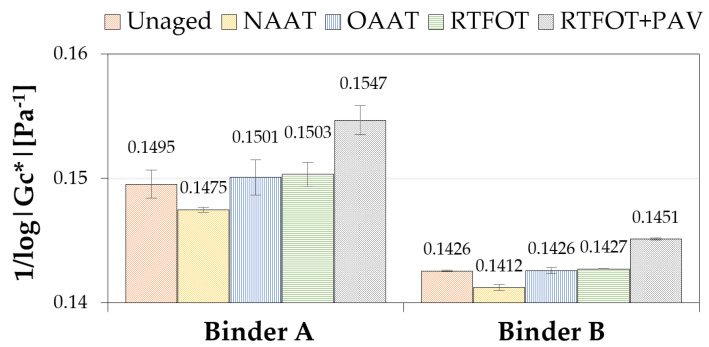
Values of the inverse of log crossover modulus for binders A and B.

**Figure 6 materials-13-04438-f006:**
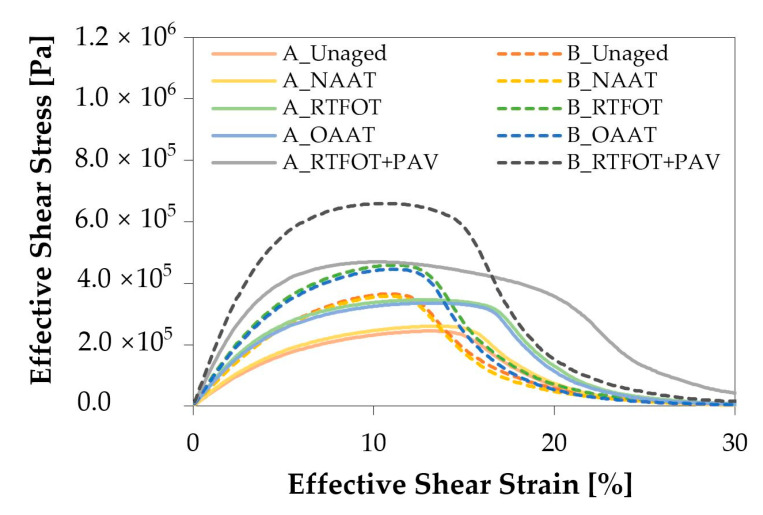
Shear stress (Pa) versus shear strain (%).

**Figure 7 materials-13-04438-f007:**
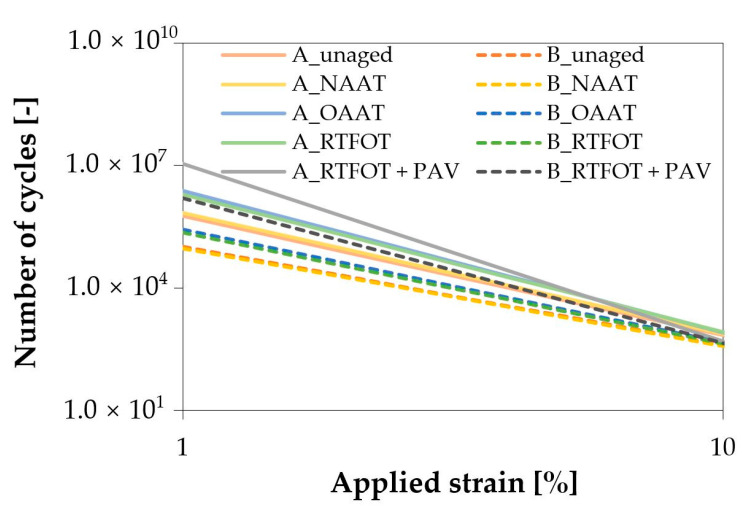
Fatigue life of binders A and B (at different aging conditions).

**Figure 8 materials-13-04438-f008:**
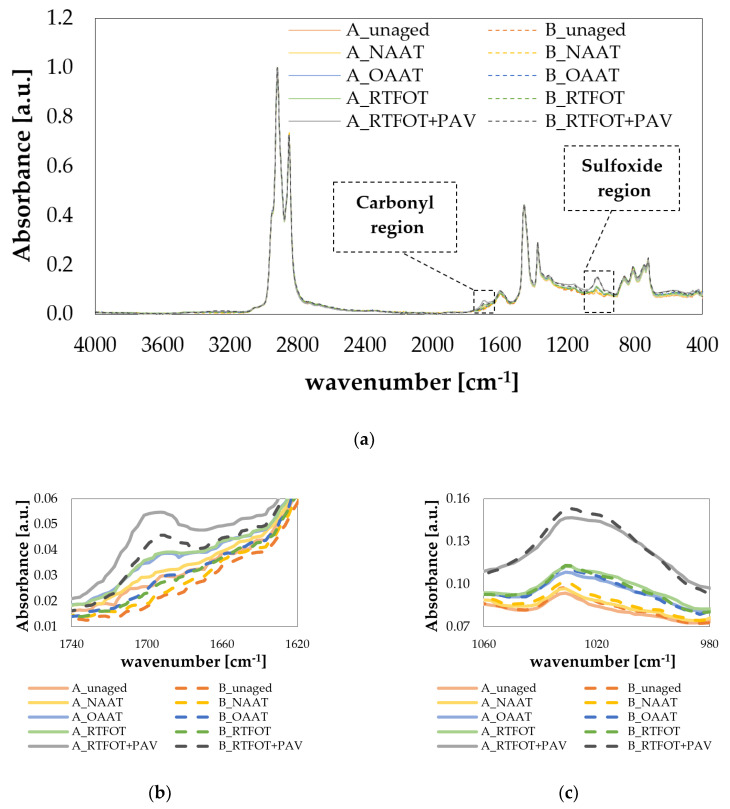
FTIR spectra: (**a**) complete spectra, (**b**) carbonyl region, and (**c**) sulfoxide region.

**Figure 9 materials-13-04438-f009:**
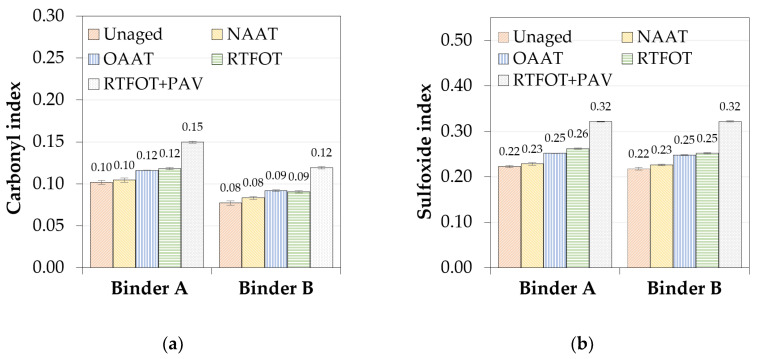
Aging indices: (**a**) carbonyl and (**b**) sulfoxides.

**Figure 10 materials-13-04438-f010:**
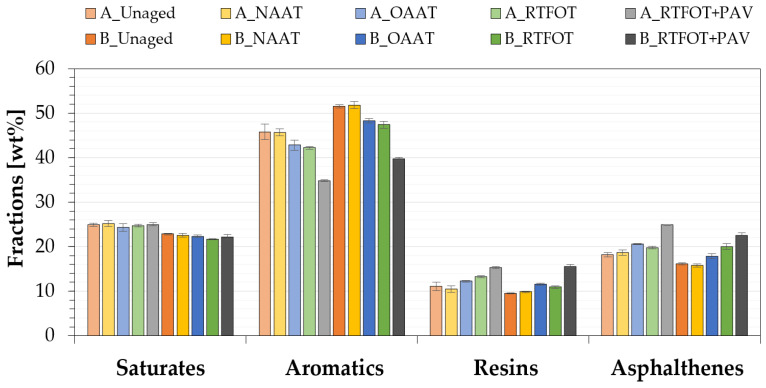
SARA fractions percentages at different aging states.

**Figure 11 materials-13-04438-f011:**
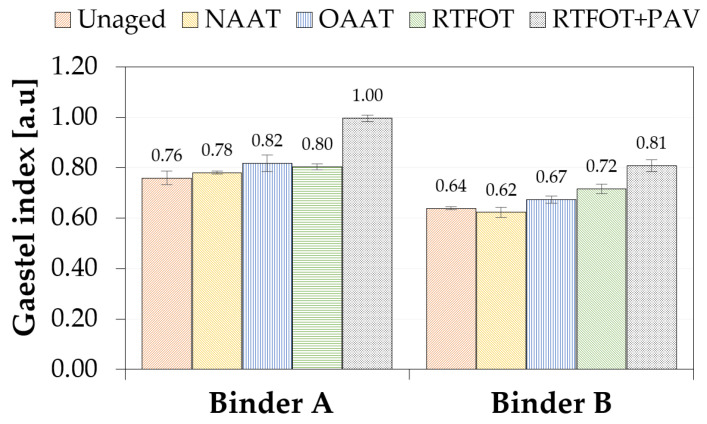
Gaestel Index.

**Table 1 materials-13-04438-t001:** Characteristics of the unmodified asphalt binders.

Binder Properties	Units	Binder A	Binder B	Specifications
Performance grade (PG)	–	64–22	64–28	–
Penetration (25 °C)	0.1 mm	79	88	EN 1426 [19]
Softening point	°C	47.2	45.0	EN 1427 [20]
Density (15 °C)	g/cm^3^	1.02	1.02	–

**Table 2 materials-13-04438-t002:** Codification used for each asphalt binder considered.

Asphalt Binder Aging State	Binder A	Binder B
Unaged binder	A_Unaged	B_Unaged
NAAT aged binder	A_NAAT	B_NAAT
OAAT aged binder	A_OAAT	B_OAAT
RTFOT aged binder	A_RTFOT	B_RTFOT
RTFOT + PAV aged binder	A_RTFOT + PAV	B_RTFOT + PAV

**Table 3 materials-13-04438-t003:** Carbonyl and sulfoxide indices integration limits.

Structural Group	Upper Wave Number Limit (cm^−1^)	Lower Wave Number Limit (cm^−1^)
Carbonyl	1755	1640
Sulfoxide	1060	984
Reference (aliphatic)	1525	1355

**Table 4 materials-13-04438-t004:** Parameters of aging procedures (rolling thin-film oven test (RTFOT), ambient atmosphere oven aging test (OAAT), and nitrogen atmosphere oven aging test (NAAT)).

Procedure	RTFOT	OAAT	NAAT
Temperature (°C)	163	163	163
Thin-film thickness (mm)	1.25	1.90	1.90
Duration (minutes)	75	240	240
Aging atmosphere	Ambient air	Ambient air	Nitrogen
Sample is submitted to continuous movement	Yes	No	No

**Table 5 materials-13-04438-t005:** Linear amplitude sweep (LAS) parameters.

Binder	Parameter A	Parameter B
A_Unaged	5.7 × 10^5^	−2.91
A_NAAT	7.0 × 10^5^	−2.97
A_RTFOT	2.4 × 10^6^	−3.47
A_OAAT	1.9 × 10^6^	−3.36
A_RTFOT + PAV	1.1 × 10^7^	−4.35
B_Unaged	1.0 × 10^5^	−2.42
B_NAAT	9.4 × 10^4^	−2.39
B_RTFOT	2.7 × 10^5^	−2.76
B_OAAT	2.3 × 10^5^	−2.73
B_RTFOT + PAV	1.6 × 10^6^	−3.56

**Table 6 materials-13-04438-t006:** Summary of the main bands and respective assignments.

Wavenumber (cm^−1^)	Assignations
2919	ν CH_2_ (aliphatic)
2850	ν CH_2_ (aliphatic)
1693	ν C=O (carbonyl)
1600	ν C=C (aromatic)
1456	δ CH_2_ (aliphatic)
1375	δ CH_3_ (aliphatic)
1030	ν S=O (sulfoxide)
865	ν CH (aromatic)
810	ν CH (aromatic)
745	ν CH (aromatic)
721	r CH_2_ (aliphatic)

Note: ν—stretching; δ—bending; r—rocking.

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
