# Peer review of "Effect of Thermal and Oxidative Aging on Asphalt Binders Rheology and Chemical Composition"

_materials, 2020, doi:10.3390/ma13194438_

Round 1

Reviewer 1 Report

The paper is overall good, interesting and well written. But, authors are invited to solve some of the comments below.

Oxidative aging is the main cause for hardening. What it misses in the manuscript is the main descriptors affecting the material´s aging. It seems that the main conclusion of the manuscript is to proof that NAAT does not duplicate the real aging effect while RTFOT+PAV does. It can also be argued that since binders after NAAT seems even softer than unaged binder, there is no really any type of aging of any nature when using NAAT. So please stress the overall goal of the paper.

In addition, I would recommend addressing the following points:

  • In chemical terms, aging leads first to a decrease in aromatic content and subsequent increase in resin content, together with a higher asphaltene content. Therefore, it is generally accepted that the aromatics generate resins which in turn generate asphaltenes. The saturates remain essentially unchanged. The SARA results of author seems to show the opposite with aromatic increasing after aging. Why?
  • In lines 46-48 authors mention steric hardening due to temperature. Specifically, bitumen and asphaltenes are believed to strongly aggregate at low temperature (60 °C). The Petersen model proposes oxidation caused by temperature. Results seem contradicting this point. Can authors comment on this aspect? Have authors registered any loss of volatile components?
  • The conclusion of the manuscript is that NAAT is not capable of reproducing the effective aging that asphalt occurs during its service life. One can argue that there are already RTFOT and PAV that can be used to experimentally simulate asphalt aging. Although it is extremely difficult to account for a complete picture of chemical reactions during oxidation, it is possible to evaluate the chemical reactivity of the molecules. Such a thing can be done by using molecular modelling and computational chemistry. These tools would be beneficial to really proof the nature of oxidation/aging.
  • In Grosseger et al. 2018, the main conclusion is that “by conducting the RTFOT with nitrogen instead of air, it was observed that the spectra showed an aged state by a general decrease in intensity compared to the signal of the base bitumen(..) this results suggest that production under nitrogen can not prohibit ageing, but might prolongate lifetime”. It sounds very similar to the conclusion this manuscript. Maybe a multi-scale approach would be more conclusive?
  • Oxidation is the main factor for the hardening of the asphalt binders. In a multiscale analysis, macroscopic properties can be experimentally studied under oxidative conditions, and then chemical reactivity can studied at a molecular level.

Table 4 does not add any relevant information since those data are very well known.

The legend of the spectra are difficult to read since the colour is the same.

Line 315 “suffer” might not be the proper verb. Maybe, undergo?

Line 93 a Reference is missing. Same for line 134.

Figure 3e and 3f are very difficult to read.

There are some sentences, which have multiple references. It would be better to mention why using both and not just one.

I would suggest enlightening the goal and the scope of the paper. Maybe a flow chart would be helpful too.

Author Response

Point 1: The paper is overall good, interesting and well written. But, authors are invited to solve some of the comments below.

Response 1: We appreciate the positive feedback from the reviewer 1.

Point 2: Oxidative aging is the main cause for hardening. What it misses in the manuscript is the main descriptors affecting the material´s aging. It seems that the main conclusion of the manuscript is to proof that NAAT does not duplicate the real aging effect while RTFOT+PAV does. It can also be argued that since binders after NAAT seems even softer than unaged binder, there is no really any type of aging of any nature when using NAAT. So please stress the overall goal of the paper. In addition, I would recommend addressing the following points:

Response 2: As suggested by the reviewer, we have stressed the overall goal of the paper introduction. However, it should be noted that the authors do not intent to provide a more realistic aging procedure (than the RTFOT+PAV) but intent to investigate the respective impact of the ageing atmosphere on the asphalt binder aging mechanism. However, changes have been made in the manuscript - Please see Page 2, lines 80 to 88.

Point 3: In chemical terms, aging leads first to a decrease in aromatic content and subsequent increase in resin content, together with a higher asphaltene content. Therefore, it is generally accepted that the aromatics generate resins which in turn generate asphaltenes. The saturates remain essentially unchanged. The SARA results of author seems to show the opposite with aromatic increasing after aging. Why?

Response 3: Thank you very much for the comment. However, we invite the reviewer to review Figure 10 (Page 15 on the manuscript) and Table 1 (below) to verify that our results showed a reduction (not an increase) in the percentage of the aromatic fraction after aging (OAAT; RTFOT and RTFOT + PAV), for both binders (A and B).

In addition, there was no significant variation (increase or decrease) in the aromatic fraction after the NAAT aging procedure. This finding indicates that there is no change in the asphalt binders composition after NAAT aging procedure.

Table 1. Aromatics factions at different aging conditions (Binders A and B)

Aging condition

Aromatics (%)

Binder A

Binder B

Unaged

45.8

51.5

NAAT

45.7

51.8

OAAT

42.8

48.2

RTFOT

42.2

47.4

RTFOT+PAV

34.8

39.7

However, the authors would like to emphasize that an overall polarity shift is occurring during aging. Hence, one cannot say that aromatics will turn into resins and resins into asphaltenes. It could be possible that some aromatics will oxidize to such extent that they are eluted within the asphaltenes. This depends on the aging conditions and the binders aging susceptibility.

Point 4: In lines 46-48 authors mention steric hardening due to temperature. Specifically, bitumen and asphaltenes are believed to strongly aggregate at low temperature (60 °C). The Petersen model proposes oxidation caused by temperature. Results seem contradicting this point. Can authors comment on this aspect? Have authors registered any loss of volatile components?

Response 4: You are correct, according to Petersen model the temperature contributes significantly to the increase of viscosity of the asphalt binders during the oxidation. In fact, according to Petersen (1993) the asphaltenes and polar aromatics molecules tend to associate during the oxidative aging, at low temperature (60 ° C). The same trendy of association of reactive polar molecules (asphaltenes and polar aromatics molecules) was observed at high temperatures (163 ° C). However, the increase of viscosity due to the association of reactive polar molecules depends on the temperature. At low temperatures, the thermal energy of the system limits the viscosity increasing of the material (to a limiting value). On the other hand, at high temperatures, the high thermal energy avoids the immobilization of the polar molecules which allows the continuing of oxidation. Moreover, at high temperatures, there is a linear relationship between viscosity and oxidative aging time.

Therefore, you are correct. It would be expected that at high temperatures the asphalt binder would oxidize more and increase its viscosity. However, this did not occur in the case of NAAT, as the interaction of the asphalt binders with oxygen was minimized/prevent due to the nitrogen atmosphere. Thus, we believe that our results do not contradict the Petersen model as there was no significant amount of oxygen molecules present during the NAAT, the asphalt binder was not oxidized so there was no increase in the material's stiffness, even when it was heated to high temperature (163 ° C) and for a long period (4 hours). We sincerely hope that the reviewer's inquiries about the possible contracting point to the Petersen model were clarified through the comments above.

Yes, the mass change rate (MCR) during the NAAT and OAAT for both binders (A and B) was recorded. The obtained values were calculated according to equation (1) (bellow) and the values obtained are presented in Figure 1:

(1)

Where M1 and M0 are the mass of the binders after and before aging, respectively.

(a)

(b)

Figure 1. Mass change rate (after NAAT and OAAT aging procedure): (a) Binder A; (b) Binder B.

For both asphalt binders it was observed a higher decrease of the binder mass after the NAAT when compared to the decrease after OAAT aging procedure. The higher decrease of the material mass during NAAT can be correlated with the volatilization of light components remaining in the fresh asphalt binder which was facilitated by the high temperatures (163 °C). In addition, the rheological results showed that the volatilization of lighter compounds during NAAT did not result in the increase of the material stiffness. One possible explanation is the low amount of volatiles present in the studied asphalt binders. Finally, the MCR values after OAAT indicate the loss of mass due to the volatilization of lighter compounds was offset by the oxidation of the material.

The authors recognized the relevance of this topic and for this reason add information about the loss of mass during NAAT and OAAT in the manuscript (Page 10, lines 318 to 333).

Petersen, J. Claine. "Asphalt oxidation--ah overview including a new model for oxidation proposing that physicochemical factors dominate the oxidation kinetics." Fuel science & technology international 11.1 (1993): 57-87.

Point 5: The conclusion of the manuscript is that NAAT is not capable of reproducing the effective aging that asphalt occurs during its service life. One can argue that there are already RTFOT and PAV that can be used to experimentally simulate asphalt aging. Although it is extremely difficult to account for a complete picture of chemical reactions during oxidation, it is possible to evaluate the chemical reactivity of the molecules. Such a thing can be done by using molecular modelling and computational chemistry. These tools would be beneficial to really proof the nature of oxidation/aging.

Response 5: The authors would like to stress that it is not the goal of this study to provide a realistic aging procedure but rather understand the impact of thermal and oxidative aging in asphalt binders. This should lay a basis for future investigations, which will intent to uncover all the complex reactions participating in the aging mechanism.

Indeed, different studies like the one conducted by Tongyan et al. (2012) and Liu et al. (2019) evidence of promising use of computational chemistry to evaluate and predict the effect of oxidative aging on the physicochemical properties of asphalt binder. This approach is promising because it allows the reliable prediction of the properties of asphalt binders after aging without requiring extensive laboratory work. Therefore, such a methodology will be considered by our research group in future studies to validate hypotheses regarding the effect of oxidation on hardening to asphalt binders. However, before we implement such a computational approach, we want to understand the impact of the specific parameters on asphalt binder aging.

Tongyan, Pan, Lu Yang, and Wang Zhaoyang. "Development of an atomistic-based chemophysical environment for modelling asphalt oxidation." Polymer degradation and stability 97.11 (2012): 2331-2339.

Liu, Fang, Zhidong Zhou, and Yu Wang. "Predict the rheological properties of aged asphalt binders using a universal kinetic model." Construction and Building Materials 195 (2019): 283-291.

Point 6: In Grosseger et al. 2018, the main conclusion is that “by conducting the RTFOT with nitrogen instead of air, it was observed that the spectra showed an aged state by a general decrease in intensity compared to the signal of the base bitumen(..) this results suggest that production under nitrogen can not prohibit ageing, but might prolongate lifetime”. It sounds very similar to the conclusion this manuscript. Maybe a multi-scale approach would be more conclusive?

Response 6: Yes, a multi-scale approach considering the aging of different scales (asphalt binder, mastic asphalt mixtures and asphalt mixtures) under nitrogen at different temperatures (60 ° C and 163 ° C) will allow a better understanding of the temperature effect on the hardening of the asphalt binders. This approach allows us to evaluate the effects of temperature together with the effect of other important factors (for example asphalt mixture air void content and aggregates characteristics) on the hardening of asphalt binders. Therefore, the multi-scale approach is one of the future studies that will be developed by our research group.

At the moment our group is focusing on the development of devices that allow the total elimination of oxygen, thus avoiding the oxidation of the material and allowing to separately evaluate the effect of temperature on asphalt binders. However, this is not a trivial task. In fact, in the study of Grosseger et al. (2018), it was mentioned that the geometry of the RTFOT oven must not have allowed the complete elimination of oxygen, thus allowing the remaining oxygen molecules to react with the asphalt binder.

Point 7: Oxidation is the main factor for the hardening of the asphalt binders. In a multiscale analysis, macroscopic properties can be experimentally studied under oxidative conditions, and then chemical reactivity can studied at a molecular level.

Response 7: Thank you very much for the valuable suggestion of approach for future works. Indeed, one of the main goal of our research group is to understand the oxidation mechanisms of asphalt binders. Therefore, in future works, the asphalt binders will be aged considering different temperatures and reactive oxygen species (ROS). The macroscopic properties will be accessed through rheological and chemical characterization while chemical reactivity will be assessed using kinetic models and microscopy techniques.

Point 8: Table 4 does not add any relevant information since those data are very well known.

Response 8: We would like to thank you for your suggestion to improve the assertiveness of the manuscript. However, we believe that table 4 (Page 8) allows the reader to easily compare the parameters (temperature, film thickness, duration, aging atmosphere, and type of agitation) considered in the proposed methods (NAAT and OAAT) and the RTFOT aging procedure. We hope that the reviewer understands our choice and justification.

Point 9: The legend of the spectra are difficult to read since the colour is the same.

Response 9: We improved the image quality and changed the colours to facilitate the reading of the legend. Please see the changes on the manuscript on page 13.

Point 10: Line 315 “suffer” might not be the proper verb. Maybe, undergo?

Response 10: You are correct, the term undergo is more appropriate. Please see the change from the verb "suffered" to "undergone", page 10 line 339.

Point 11: Line 93 a Reference is missing. Same for line 134.

Response 11: The missing reference issues were corrected. Please see pages 3 and 4.

Point 12: Figure 3e and 3f are very difficult to read.

Response 12: In order to facilitate the visualization of figures 3e and 3f, their quality and size have been changed. Please see page 9.

Point 13: There are some sentences, which have multiple references. It would be better to mention why using both and not just one.

Response 13: The choice of several references aims to ensure that the information provided was mentioned by different authors. We hope that the reviewer understands our approach in order to assure that the statements done on the paper were mainly based on different relevant studies.

Point 14: I would suggest enlightening the goal and the scope of the paper. Maybe a flow chart would be helpful too.

Response 14: As suggested by the reviewer, we have stressed the goal of the paper introduction. Please see Page 2, lines 80 to 88.

Reviewer 2 Report

This study proposed two novel two thin film aging test procedures to investigate the effect of thermal and oxidative aging on asphalt binders. It is a very interesting study and well written paper, but there are spaces to improve the manuscript.

  1. The key words of the article can include the chemical composition.
  2. There appears some typing error at Page 3 line 93.
  3. Please avoid using the term asphalt binders. I see only unmodified asphalt is considered in your study, however, modified bitumen is also widespread in engineering fields. Some exemplary studies have shown that the aging behavior of modified asphalt is different from base asphalt, such as the research Doi: https://doi.org/10.1016/j.conbuildmat.2019.117732. So I suggest you use unmodified asphalt binders in place of asphalt binders in your introduction.
  4. Why you give a reference group (aliphatic) in Page 6 line 207, please explain.
  5. Page 8 line 282, the title of Figure 3(e) and (f) are the same? Title does not reflect content adequately.
  6. The Figure 3 in the article are not very clear, especially the curves in the pictures.
  7. Page 11 line 358, you say the band area around the peaks at 1600cm-1 was observed in your study, but no discussion about the variation of this aromatic function under different aging procedure.
  8. Why you list the main bands and respective assignments in Table 4, the reasons should be explained in detail in the manuscript.
  9. Page 13 line 401, the reasons why “the asphalt binder stiffness must be assessed from the total sum of the resins and asphaltenes fractions’ should be explained in detail in the manuscript.
  10. Please check the number of the tables.

Author Response

Point 1: This study proposed two novel two thin film aging test procedures to investigate the effect of thermal and oxidative aging on asphalt binders. It is a very interesting study and well written paper, but there are spaces to improve the manuscript.

Response 1: We thank the reviewer for their careful reading of the manuscript and constructive

suggestions.

Point 2: The key words of the article can include the chemical composition.

Response 2: We agree that the keyword “chemical composition” fits well with the paper context. It was included in the keywords (please see Page 1, lines 26 and 27). We believe this change improves the visibility of the paper among our target audience. Thank the reviewer for the valuable suggestion.

Point 3: There appears some typing error at Page 3 line 93.

Response 3: Thank you very much for the comment. The missing reference issues were corrected. Please see pages 3 and 4.

Point 4: Please avoid using the term asphalt binders. I see only unmodified asphalt is considered in your study, however, modified bitumen is also widespread in engineering fields. Some exemplary studies have shown that the aging behavior of modified asphalt is different from base asphalt, such as the research

Doi: https://doi.org/10.1016/j.conbuildmat.2019.117732. So I suggest you use unmodified asphalt binders in place of asphalt binders in your introduction.

Response 4: We thank the reviewer for identifying this potential ambiguity and for the suggestion given. We agree that we should be more cautious with the generalization of the term asphalt binder. Since different relevant studies (as the one referenced by the reviewer) indicate that the mechanisms of degradation observed for unmodified asphalt binders and modified asphalt binders are different. This study intents to investigate the fundamental impact of oxygen during aging on the pure bitumen level. By adding polymer to the equation, an entirely new level of complexity is added, which will be even harder to understand. Hence, this first approach was mainly focused on unmodified binders. However, changes were made to the manuscript on page 2, lines 82, 86, 88 and 90.

Point 5: Why you give a reference group (aliphatic) in Page 6 line 207, please explain.

Response 5: We appreciate the detailed investigation of our approach to the determination of the carbonyl and sulfoxide indices. The reference group (aliphatic) was considered on the calculation of the indices in order to allow the comparison between samples with different film thicknesses. The aliphatic group was chosen since its structure is not affected by the ageing (Hofko et al. (2017)). A similar calculation approach can be found in the published paper of Mirwald et al. (2020). 

In addition, the choice of determining the best approach for the calculation of carbonyl and sulfoxide indices is still an ongoing discussion in our working groups. Since, it is very important to assure that those indices allow the comparison of the oxidation degree of asphalt binders at different aging conditions, regardless of the thickness of the sample considered.  

We hope that the reviewer understands our choice and justification.

Hofko, Bernhard, et al. "Repeatability and sensitivity of FTIR ATR spectral analysis methods for bituminous binders." Materials and Structures 50.3 (2017): 187.

Mirwald, Johannes, et al. "Impact of reactive oxygen species on bitumen aging–The Viennese binder aging method." Construction and Building Materials 257 (2020): 119495.

Point 6: Page 8 line 282, the title of Figure 3(e) and (f) are the same? Title does not reflect content adequately.

Response 6: We thank you very much for the comment. In fact, the caption for figure 3 (f) was wrong. Please see the correction on the page 9, line 315 and 316.

Point 7: The Figure 3 in the article are not very clear, especially the curves in the pictures.

Response 7: As suggested by the reviewer, we have improved the quality and size of figures 3e and 3f. Please see the correction on the page 9.

Point 8: Page 11 line 358, you say the band area around the peaks at 1600cm-1 was observed in your study, but no discussion about the variation of this aromatic function under different aging procedure.

Response 8:

We decided for not adding the aromatic indices on the manuscripts due to the little availability of studies that correlate oxidative aging with the increase in the peak 1600 cm-1 region. In fact, there are some relevant studies (Lamontagne et al. (2001) and Nivitha et al. (2016)) which have presented aromatic indices analysis of FTIR results. However, most of the published studies on ATR FTIR spectroscopic analysis for asphalt binders focus only on the carbonyl and sulfoxide indices as they represent the two major functional groups that are formed upon aging. Additionally, such a shift in the entire fingerprint area can be linked to an increase in the polarity of the material (Mirwald et al. (2020). Hence, we believe that further studies are needed to assure the effectiveness of aromatic indices on the evaluation of the effect of oxidation of asphalt binders.

For this reason, we have not decided not to present the results of aromatic indices in the manuscript. However, this is a topic that has been frequently debated in our research group and will be better addressed in our future researches.

We hope that the reviewer understands our concerns and justification.

Lamontagne, J., et al. "Comparison by Fourier transform infrared (FTIR) spectroscopy of different ageing techniques: application to road bitumens." Fuel 80.4 (2001): 483-488.

Nivitha, M. R., Edamana Prasad, and J. M. Krishnan. "Ageing in modified bitumen using FTIR spectroscopy." International Journal of Pavement Engineering 17.7 (2016): 565-577.

  1. Mirwald, S. Werkovits, I. Camargo, D. Maschauer, B. Hofko, H. Grothe: "Investigating Bitumen Long-Term-Ageing in the Laboratory by Spectroscopic Analysis of the SARA Fractions"; Construction and Building Materials, (accepted)

Point 9: Why you list the main bands and respective assignments in Table 4, the reasons should be explained in detail in the manuscript.

Response 9: Changes to the manuscript were made, explaining the information presented in Table 6. Please, see the inserted text on Page 12, lines 395 to 398.

Point 10: Page 13 line 401, the reasons why “the asphalt binder stiffness must be assessed from the total sum of the resins and asphaltenes fractions’ should be explained in detail in the manuscript.

Response 10: Adaptations have been made in the manuscript. Please see page 14 (lines 429 to 435) for changes made as suggested by the reviewer.

Point 11: Please check the number of the tables.

Response 11: In fact, the number of tables was incorrect. We corrected this issue along with the whole manuscript.

Reviewer 3 Report

The manuscript is focused on the assessment of the aging of bitumen binders and the effect of thermal and oxidative aging on binder rheology properties and changes in composition due to aging using new test methods. This manuscript is well written and easily followed. The following suggestions are important to improve the quality of a paper:

  1. In the introduction at the ending part, the authors summarize the results, but this part of the paper should focus on the reason for the research goal.
  2. Presented research of aging was not performed on two distinct types of asphalt/bitumen binders, but on the same type of paving grade bitumen 70/100. Different types (PMB) and/or different gradations (50/70) of bitumen are important to demonstrate the suitability of the new method (NAAT).
  3. Chapter 3 Results and discussion; although the manuscript focuses on rheological properties it is necessary to supplement the results of a change in Penetration value, Softening Point value, and mass of aged bitumen samples. This is important for evaluating the effects of aging on bitumen (especially if new aging conditions NAAT are to be assessed).
  4. The values of Cross Modulus (chapter 3.1) of origin and short term aged bitumen (RTFOT and OAAT) are the same, otherwise thermal aging without oxygen changes the susceptibility of bitumen, unlike other results. Needs to be explained.
  5. Chapter 3.3; results of FTIR absorbance spectra were analysed in terms of carbonyl and sulfoxide chemical compounds. The NAAT aging test exposes bitumen the nitrogen and thus nitrogen-containing compounds should be analysed.
  6. The high values of the Gaestel index from SARA analyse (chapter 3.4) indicate colloidal unstable bitumen, standard value are between 0.22 and 0.50. Needs to be explained.
  7. Reference source not found in chapter 2.1 and 2.2.
  8. Figure 3; add reference temperature.
  9. Figure 4 does not express correlation.
  10. Figure 4 check the value 1/Gc* (if the Cross Modulus value of 106 to 107 Pa).
  11. Figure 6 does not show fatigue life at 5% strain.

Author Response

Point 1:  The manuscript is focused on the assessment of the aging of bitumen binders and the effect of thermal and oxidative aging on binder rheology properties and changes in composition due to aging using new test methods. This manuscript is well written and easily followed. The following suggestions are important to improve the quality of a paper:

Response 1: We would like to thank the reviewer for positive feedback and constructive comments.

Point 2: In the introduction at the ending part, the authors summarize the results, but this part of the paper should focus on the reason for the research goal.

Response 2: Thank you very much for the recommendations. As suggested by the reviewer, we have stressed the overall goal of the paper introduction. Please see Page 2, lines 80 to 88.

Point 3: Presented research of aging was not performed on two distinct types of asphalt/bitumen binders, but on the same type of paving grade bitumen 70/100. Different types (PMB) and/or different gradations (50/70) of bitumen are important to demonstrate the suitability of the new method (NAAT).

Response 3: We thank the reviewer for highlighting this important aspect to guarantee the applicability of the new methods (NAAT and OAAT) for different types of unmodified and modified asphalt binders. However, the main objective of this study was to verify the applicability of this method for 70/100 penetration graded asphalt binders (one of the most common asphalt binders used in Central and Western Europe for road pavements). Besides, the choice for an unmodified binder sought to eliminate variables that could affect the thermal aging susceptibility of the asphalt binder, which would make it difficult to understand the results. However, it should be noted that the two asphalt binders considered in this study come from different refineries and crude oil origins. Therefore, as shown in the manuscript, both binders differ in terms of rheological behavior and chemical composition.

Since the NAAT and OAAT methods were effective for the two unmodified binders considered in this study, we will surely consider different types of binders in future research.

Point 4: Chapter 3 Results and discussion; although the manuscript focuses on rheological properties it is necessary to supplement the results of a change in Penetration value, Softening Point value, and mass of aged bitumen samples. This is important for evaluating the effects of aging on bitumen (especially if new aging conditions NAAT are to be assessed).

Response 4: This is an important point. We decided to focus on rheological analysis because in this approach it is possible to consider dynamic loading and a wide temperature range. Which allows us to better represent the conditions that the material is subject to in the field. However, in the future work, we hope to show a detailed Penetration value, Softening Point value, and mass of all aged asphalt binders.

Point 5: The values of Cross Modulus (chapter 3.1) of origin and short term aged bitumen (RTFOT and OAAT) are the same, otherwise thermal aging without oxygen changes the susceptibility of bitumen, unlike other results. Needs to be explained.

Response 5:  We decided to present the values of the inverse of the log crossover modulus (1/log|Gc*|) since some studies (Farrar et al. (2013) and Liu and Wen (2015)) showed a good correlation between this parameter and carbonyl content. Indeed, our results also indicated a good correlation between both parameters (please see Figure 1).

Figure 1. Correlation of carbonyl index and (1/log|Gc*|) of binders A and B.

As pointed out by the reviewer it was expected that the values of (1/log|Gc*|) registered by asphalt binders at unaged and short-term aging condition were significantly different. However, after rechecking the calculation we confirmed that the obtained values of (1/log|Gc*|) for unaged condition were very similar to short-term conditions.

In order to compare the results obtained in our studies, we considered the published results of Jing et al. (2019) (Figure 2, bellow) to calculate the values of (1/log|Gc*|) at unaged; RTFOT and RTFOT+PAV condition. The calculated values are given in Table 1 (bellow):

Figure 2. Crossover modulus published by Jing et al. (2019).

Table 1. Calculated values of of (1/log|Gc*|)

Aging condition

|Gc*| [MPa] (*)

1/log|Gc*| [Pa-1] (**)

Unaged

18.8

0.137

RTFOT

17.5

0.138

RTFOT+PAV

4.9

0.149

Note: (*) value obtained from the Figure 2; (**) Calculated values.

The calculated values of (1/log|Gc*|) (Table 1) also showed significative similarity between the values at unaged and RTFOT aged condition. Given the similarity on the values of (1/log|Gc*|) found in both studies, we hypothesize that this parameter is not suitable to differentiate unaged from short-term aged samples.

Farrar, Michael J., et al. "Evolution of the crossover modulus with oxidative aging: method to estimate change in viscoelastic properties of asphalt binder with time and depth on the road." Transportation research record 2370.1 (2013): 76-83.

Liu, Fang, and Haifang Wen. "Prediction of rheological and damage properties of asphalt binders that result from oxidative aging." Transportation Research Record 2505.1 (2015): 92-98.

Jing, Ruxin, et al. "Ageing effect on chemo-mechanics of bitumen." Road Materials and Pavement Design (2019): 1-16.

Point 6: Chapter 3.3; results of FTIR absorbance spectra were analysed in terms of carbonyl and sulfoxide chemical compounds. The NAAT aging test exposes bitumen the nitrogen and thus nitrogen-containing compounds should be analysed.

Response 6: A visual analysis was made to compare the spectrum of a given asphalt binder (A or B) in the unaged condition and after NAAT aging procedure. For both asphalt binders, no significative difference was observed between the spectra at unaged and NAAT aged conditions. Since nitrogen (N2) is an inert gas with bond-dissociation energy of 945 kJ/mol, it is not prone to react with anything, as the activation barrier is very high and unlikely to be overcome at the giver temperatures and reaction conditions. Thus, it is believed that no nitrogen compound was formed during the NAAT procedure. However, it is a relevant topic that will be evaluated in more detail by different chemical analyses in future work (as there are also other nitrogen-containing reactive species in our atmosphere, which can contribute to aging). Given the relevance of the matter, we inserted on the manuscript (please check Page 12, lines 391 to 394) a sentence indicating that future chemical analyses should be conducted to evaluate the formation of nitrogen compounds during NAAT aging procedure.

Point 7: The high values of the Gaestel index from SARA analyse (chapter 3.4) indicate colloidal unstable bitumen, standard value are between 0.22 and 0.50. Needs to be explained.

Response 7: We thank the reviewer for bringing up this important topic that can be further explored in the manuscript. The possible explanation for the high values of Gaestel index (Ic) presented is due to the methodology used for the SARA fractionation. According to the semi-quantitative analysis presented in the study by Sakib et al. (2019), this new methodology presented a higher yield of asphaltenes when compared to the values obtained by ASTM D4124 standard. One of the possible explanations for the difference between the percentage of asphaltenes obtained by both procedures is the pore size of the filter. In the procedure developed by Sakib et al. (2019) the PTFE syringe filter has a much smaller pore size (0.2 µm) than that the glass filter (approximately 10 µm) considered in ASTM D 4124.

Given the relevance of the topic, a sentence was added to the manuscript to explain the possible reasons for the high values of Ic. Please see Page 15, lines 464 to 472.

Sakib, Nazmus, and Amit Bhasin. "Measuring polarity-based distributions (SARA) of bitumen using simplified chromatographic techniques." International Journal of Pavement Engineering 20.12 (2019): 1371-1384.

Point 8: Reference source not found in chapter 2.1 and 2.2.

Response 8: The missing reference issues were corrected. Please see pages 3 and 4.

Point 9: Figure 3; add reference temperature.

Response 9: We added the reference temperature in Figure 3 (a) to (f). Please see Page 9.

Point 10: Figure 4 does not express correlation.

Response 10:We have changed the title of Figure 5. Please Page 10 (Figure 5).

Point 11: Figure 4 check the value 1/Gc* (if the Cross Modulus value of 106 to 107 Pa).

Response 11: We thank the reviewer for highlighting this important point. The correct values are from 0.14 Pa-1 to 0.16 Pa-1.

Point 12: Figure 6 does not show fatigue life at 5% strain.

Response 12:We have changed the title of Figure 7. Please Page 12, line 377.

Reviewer 4 Report

The author reported “Effect of thermal and oxidative aging on asphalt binders rheology and chemical composition”. This manuscript focuses on the influence of thermal aging and oxidative aging on asphalt hardness. The paper is not suitable to be accepted in its present form. To be acceptable the following modifications are necessary.

  1. Please address the novelty of this investigation in the “Introduction” section.
  2. The annotation for Figure 3(f) in line 281 does not match the picture.
  3. Why the 1/|Gc*| of unaged state show a difference from NAAT, but very similar with RTFOT and OAAT in Figure 4, though the curves of |G*| of unaged state and NAAT almost overlaid in Figure 3.
  4. In table 3, the absolute value of the parameter A of binder A is not always higher than binder B, which is not consistent with the conclusion in line 331. Please revise the value.
  5. Please supplement the link between colloid and Gaestel index in Figure 10.
  6. Most importantly, the assumption of NAAT is based on oxygen is the only variable between NAAT and OTTA. However all results for NAAT show that NAAT also suppresses the volatilization of light components, resulting the absence of oxidative and thermal aging during NAAT aging procedure. Thus the main conclusion that oxidative aging is the main cause for the hardening is not convincing enough. It is recommended to explain why there is no thermal aging in the nitrogen atmosphere? Or add necessary experiments to verify the author's main conclusion.

Author Response

Point 1:  The author reported “Effect of thermal and oxidative aging on asphalt binders rheology and chemical composition”. This manuscript focuses on the influence of thermal aging and oxidative aging on asphalt hardness. The paper is not suitable to be accepted in its present form. To be acceptable the following modifications are necessary.

Response 1: We thank the reviewer for constructive comments and suggestions. We hope that the quality of the manuscript is improved after the needed modifications.

Point 2: Please address the novelty of this investigation in the “Introduction” section.

Response 2: The overall goal of this study is to provide a fundamental understanding of the impact of thermal and oxidative aging atmosphere, which will ultimately help us to understand the aging mechanism occurring in the material. As suggested by the reviewer, we have stressed the overall goal and novelty of this investigation in the introduction section. Please see Page 2, lines 80 to 88.

Point 3: The annotation for Figure 3(f) in line 281 does not match the picture.

Response 3: Adaptations have been made in the manuscript. Please see the correction on page 9, line 316.

Point 4: Why the 1/|Gc*| of unaged state show a difference from NAAT, but very similar with RTFOT and OAAT in Figure 4, though the curves of |G*| of unaged state and NAAT almost overlaid in Figure 3.

Response 4: As pointed out by the reviewer it was expected that the values of (1/log|Gc*|) registered by asphalt binders at unaged and short-term aging conditions were significantly different. However, after rechecking the calculation we confirmed that the obtained values of (1/log|Gc*|) for the unaged condition were very similar to short-term conditions.

To compare the results obtained in our studies, we considered the published results of Jing et al. (2019) (Figure 1, bellow) to calculate the values of (1/log|Gc*|) at unaged; RTFOT and RTFOT+PAV condition. The calculated values are given in Table 1 (bellow):

Figure 1. Crossover modulus published by Jing et al. (2019).

Table 1. Calculated values of of (1/log|Gc*|)

Aging condition

|Gc*| [MPa] (*)

1/log|Gc*| [Pa-1] (**)

Unaged

18.8

0.137

RTFOT

17.5

0.138

RTFOT+PAV

4.9

0.149

Note: (*) value obtained from the figure (1); (**) Calculated values.

The calculated values of (1/log|Gc*|) (Table 1) also showed significative similarity between the values at unaged and RTFOT aged conditions. Given the similarity on the values of (1/log|Gc*|) found in both studies, we hypothesize that this parameter is not suitable to differentiate unaged from short-term aged samples.

Point 5: In table 3, the absolute value of the parameter A of binder A is not always higher than binder B, which is not consistent with the conclusion in line 331. Please revise the value.

Response 5: We revised the values and they are corrected. We invite the reviewer to review Table 5 (Page 11, on the manuscript) and Figure 2 and Figure 3 (bellow) to verify that our results showed that binder A presented a higher absolute value of the parameters (A and B), for the same aging condition.

Figure 2. Parameter A values for asphalt binders A and B.

Figure 3. Parameter B values (absolute value) for asphalt binders A and B.

Point 6: Please supplement the link between colloid and Gaestel index in Figure 10.

Response 6: We thank the reviewer for bringing up this important topic that must be further explored in the manuscript. Please see the sentence which correlates colloid and Gaestel index in Page 15, lines 451 to 460.

Point 7: Most importantly, the assumption of NAAT is based on oxygen is the only variable between NAAT and OTTA. However all results for NAAT show that NAAT also suppresses the volatilization of light components, resulting the absence of oxidative and thermal aging during NAAT aging procedure. Thus the main conclusion that oxidative aging is the main cause for the hardening is not convincing enough. It is recommended to explain why there is no thermal aging in the nitrogen atmosphere? Or add necessary experiments to verify the author's main conclusion.

Response 7:  We have addressed this issue by adding a section about the mass loss of the NAAT and OAAT which supports the conclusion that mass loss is indeed occurring during the NAAT, while during the OAAT no significant mass loss is reported due to the simultaneous incorporation of oxygen (see manuscript Page 10, Fig 4). An explanation to why no thermal aging occurs during the nitrogen atmosphere can be given as follows: As nitrogen (N2) is an inert gas with a bond-dissociation energy of 945 kJ/mol, it is not prone to react with anything, as the activation barrier is very high and unlikely to be overcome at the giver temperatures and reaction conditions.

We hope this provides some evidence on this matter.

Round 2

Reviewer 1 Report

The manuscript has been significantly improved and all comments have been addressed.

Reviewer 4 Report

Thanks for the reply and explanation, no further comments